# Consecutive Low Doses of Streptozotocin Induce Polycystic Ovary Syndrome Features in Mice

**DOI:** 10.3390/ijms22031299

**Published:** 2021-01-28

**Authors:** Youngjae Ryu, Yong Jin Kim, Yoon Young Kim, Jungwoo Kim, Sung Woo Kim, Hoon Kim, Seung Yup Ku

**Affiliations:** 1Biomedical Research Institute, Seoul National University Hospital, Seoul 03080, Korea; dragonkai@naver.com (Y.R.); yoonykim@snu.ac.kr (Y.Y.K.); 2Department of Obstetrics and Gynecology, Korea University College of Medicine, Seoul 02841, Korea; zinigo@gmail.com; 3Institute of Reproductive Medicine and Population, Medical Research Center, Seoul National University College of Medicine, Seoul 03080, Korea; kjw8249@gmail.com; 4Department of Obstetrics and Gynecology, Seoul National University Hospital, Seoul 03080, Korea; byulbi@naver.com (S.W.K.); obgyhoon@gmail.com (H.K.); 5Department of Obstetrics and Gynecology, Seoul National University College of Medicine, Seoul 03080, Korea

**Keywords:** polycystic ovary syndrome, streptozotocin, animal models, ovary, testosterone

## Abstract

Polycystic ovarian syndrome (PCOS) is a common reproductive endocrine disorder in reproductive-age women. Due to its various pathophysiological properties and clinical heterophenotypes, the mechanism of PCOS pathogenesis is still unclear. Several animal models have been used to study PCOS and allow the exploration of the specific mechanism underlying PCOS. We focused on streptozotocin (STZ) to develop a non-steroidal and non-diabetic PCOS model. We administered multiple STZ injections to female C57BL/6 mice (3–4 weeks old) at different concentrations: STZ-15 (15 mg/kg), STZ-30 (30 mg/kg), and STZ-60 (60 mg/kg) treatments. During the experimental period, we analyzed body weight, blood glucose levels, and estrous cycle pattern. Furthermore, five weeks after STZ administration, we examined hormone levels and the morphology of ovarian tissues. Mice in the STZ-15 group did not show differences in body weights, blood glucose level, insulin level, and insulin tolerance compared to wild-type and control groups whereas those in the STZ-60 group presented a typical diabetes phenotype. In the case of the STZ-30 group, only increased blood glucose level was observed. Total testosterone levels were significantly elevated in STZ-15 and STZ-30 groups. Luteinizing hormone (LH) and estradiol levels were not significantly changed in the STZ-treated groups. The number of ovarian antral follicles and atretic follicles significantly increased in the ovary of mice in the STZ-15 and STZ-30 groups. All STZ-treated groups manifested irregular estrus cycles. However, the patterns of estrous cycles were different between mice treated with different STZ concentrations. We found that PI3K-AKT and IRS-1 signaling in the ovary was enhanced by low doses of STZ treatment. Taken together, our finding indicates that multiple injections of STZ at low doses induce PCOS features in mice without induction of diabetes features.

## 1. Introduction

Polycystic ovary syndrome (PCOS) is one of the most common reproductive endocrine disorders in women [1]. According to Rotterdam’s criteria for PCOS diagnosis, the key features of PCOS are hyperandrogenism, polycystic ovarian morphology, and amenorrhea or oligo-menorrhea [2]. Besides, PCOS is frequently accompanied by metabolic abnormalities, such as insulin resistance, hyperinsulinemia, and hepatic diseases [1,3]. As a consequence, PCOS is a common reason for infertility and low quality of life of patients.

The phenotypes of PCOS women are diverse. Although the disease is well-defined, the degree of manifestation of key symptoms and metabolic characteristics greatly vary [4]. For example, Asian PCOS women rarely show hirsutism, obesity, and signs of diabetes compared with Western women [5,6,7]. Therefore, the pathophysiology of PCOS, including its cellular and molecular mechanisms, is still poorly understood.

To study PCOS, various animal models have been reported. This has been well-reviewed by abundant literature [8,9,10,11,12]. Among them, animals exposed prenatally or postnatally to high levels of androgens, such as dihydrotestosterone (DHT) or dehydroepiandrosterone (DHEA), have been extensively used. This approach focused on an important clinical feature of PCOS: hyperandrogenism. Indeed, these animal models successfully manifest characteristic features of PCOS and related metabolic symptoms. However, they have some limitation to provide evidence that allows the elucidation of PCOS pathogenesis, particularly, because of the variability among PCOS phenotypes, there are rare experimental models representative of PCOS as a whole so far. Therefore, there is a need to develop a PCOS animal model that is neither obese nor under steroidal hormone treatment.

Streptozotocin (STZ) is widely used to induce experimental type 1 diabetes [13], as well as to establish several other animal disease models [14,15]. It is known that oxidative stress and cellular damages are elicited by STZ injections [16]. Notably, Leaming et al. reported that STZ-treated female rodents show increased plasma testosterone levels [17]. In addition, disruption of the hypothalamic-pituitary axis and alteration of ovarian receptor expression are associated with STZ treatment [18,19]. However, there are no reports on the effects of STZ on the induction of other PCOS-related features, such as polycystic ovary morphology, so far. Similarly, molecular mechanisms associated with STZ administration in ovarian tissues have not been explored yet.

Mechanically, several reports suggest that the phosphoinositide 3-kinases (PI3K)–AKT molecular pathway is implicated in PCOS-related hyperandrogenism [20,21,22]. Accordingly, studies show that the PI3K-AKT pathway is dysregulated in animal PCOS models [23,24], and enhanced PI3K-AKT signaling is associated with ovarian dysfunctions and increased androgen synthesis [25,26,27].

Therefore, in the present work, we focused on the use of STZ to develop PCOS features in mice. Considering the action of STZ, we hypothesized that STZ administration would induce PCOS-like phenotypes in mice regardless of the onset of typical diabetes. We evaluated three major components of PCOS: elevated androgen level, polycystic ovarian morphology, and irregular estrous cycle in mice. Additionally, we explored the ovarian molecular pathway altered by STZ, focusing on IRS-1 and AKT signaling. Based on our results, we suggest that multiple injections of STZ at low doses are sufficient for developing PCOS-like features in mice.

## 2. Results

### 2.1. Multiple Low-Dose STZ Treatments Do Not Induce Typical Diabetic Profiles in Mice

To examine general metabolic alterations in response to various doses of STZ, we evaluated body weight and blood glucose levels weekly. Only the group treated with 60 mg/kg STZ showed significantly decreased body weight compared with wild type (WT) and control (CON) groups one week after the treatment (*n* = 8/group, F_20,175_ = 4.346, *p* < 0.0001, two-way ANOVA with Tukey’s post-hoc comparison) (Figure 1A). As expected, the blood glucose levels in the STZ-60 group were significantly elevated compared with the other groups during the observation period (*n* = 7–8/group, F_16,136_ = 0.840, *p* < 0.0001, two-way ANOVA with Tukey’s post-hoc comparison) (Figure 1D). Blood glucose levels in the STZ-30 group were significantly increased compared with the CON group at 2, 4, and 5 weeks after the treatment (Figure 1B). However, mice in the STZ-15 group did not show any significantly elevated blood glucose levels compared with the WT or CON group during the test period.

Subsequently, to explore insulin levels and insulin tolerance, which are indicators of diabetes, we measured the fasting insulin level of each group 5 weeks after the treatment (Figure 1C). The STZ-60 group showed significantly decreased fasting insulin levels compared with WT and CON groups (*n* = 6–7/group, F_4,29_ = 3.121, *p* = 0.0146, one-way ANOVA with Tukey’s post-hoc comparison), whereas the STZ-15 and STZ-30 groups were not different from WT and CON mice.

Regarding the result for insulin tolerance tests (ITT), the STZ-60 group did not present decreased blood glucose compared with the other groups at 15 min and 30 min after insulin administration (*n* = 7–8/group, F_24,198_ = 4.773, *p* < 0.0001, two-way ANOVA with Tukey’s post-hoc comparison). In the case of the HOMA-IR index, a significant difference was observed only in the STZ-60 group (*n* = 6–7/group, F_4,29_ = 2.668, *p* < 0.0001, one-way ANOVA with Tukey’s post-hoc comparison) (Figure 1E). In summary, multiple low-dose STZ injections, especially at 15 mg/kg, did not result in the typical diabetes phenotype in mice.

### 2.2. Low Doses of STZ Induces Hyperandrogenism but Do Not Increase LH or E2 Levels

To explore the hormonal status of the STZ-treated mice, especially whether animals suffer from hyperandrogenism or not, we measured total testosterone, LH, and E2 levels at the end-point of the experiment. Mice in the STZ-15 and STZ-30 groups had significantly elevated testosterone levels compared with the WT and CON groups, whereas the STZ-60 group did not (*n* = 7–8/group, F_4,34_ = 9.454, *p* < 0.0001, one-way ANOVA with Tukey’s post-hoc comparison) (Figure 2A). Luteinizing hormone stimulates the synthesis of androgens by ovarian theca cells, whose over-activity is highly related to hyperandrogenism [3,23]. In the case of LH levels, there were no statistical differences between groups except for the level in the STZ-30 group, which significantly decreased compared with the WT group (*n* = 6–7/group, F_4,29_ = 3.684, *p* = 0.0152, one-way ANOVA with Tukey’s post-hoc comparison) (Figure 2B). Mice E2 levels did not differ between groups (*n* = 4–6/group, F_4,23_ = 2.383, *p* = 0.0811, one-way ANOVA with Tukey’s post-hoc comparison) (Figure 2C).

### 2.3. Polycystic Ovaries in Mice Treated with Low Doses of STZ

Next, we performed histological evaluations after the end of the experiments. First, we found that ovarian weights were not significantly different between experimental groups (*n* = 7–8/group, F_4,34_ = 1.798, *p* = 0.152, one-way ANOVA with Tukey’s post-hoc comparison) (Figure 3A). This suggests that neither severe atrophy nor hyperplasia occurred. However, mice ovary phenotypes were affected by STZ injections. The appearance of ovaries in STZ-treated groups was atypical: They had twisted and polycystic morphologies compared with the ovaries of mice in the WT and CON groups (Figure 3B). Among the treated groups, the number of antral follicles (per ovary) of STZ-15 and STZ-30 groups significantly increased, which represents a PCOS feature (*n* = 7–8/group, F_4,33_ = 8.135, *p* < 0.0001, one-way ANOVA with Tukey’s post-hoc comparison) (Figure 3C). Furthermore, the number of atretic follicles (per ovary) significantly increased in STZ-treated groups compared with control mice (*n* = 7–8/group, F_4,33_ = 10.96, *p* < 0.0001, one-way ANOVA with Tukey’s post-hoc comparison) (Figure 3D). The number of corpora lutea (per ovary) in the STZ-60 group significantly decreased, whereas that in the STZ-15 group, slightly increased, compared with the CON group (*n* = 7–8/group, F_4,33_ = 7.703, *p* = 0.0002, one-way ANOVA with Tukey’s post-hoc comparison) (Figure 3E).

In addition to the number of follicles and corpora lutea, we measured the diameters of follicles and ovaries, as well as theca cell layer thickness. The hypertrophy of this layer is commonly associated with PCOS in animal models and humans [11,28]. In contrast to the absence of changes in their weight, the mean diameter of ovaries significantly increased in the STZ-15 group compared with control mice (*n* = 7–8 of each group, F_4,33_ = 4.799, *p* = 0.0037, one-way ANOVA with Tukey’s post-hoc comparison) (Figure 4A). The mean diameter of mature follicles (antral follicles), an ovarian follicle during a certain latter stage of folliculogenesis with formed fluid-filled cavity adjacent to the oocyte called the antrum, was not significantly different between groups (*n* = 6–8/group, F_4,31_ = 0.598, *p* = 0.666, one-way ANOVA with Tukey’s post-hoc comparison) (Figure 4B). However, corpora lutea were significantly enlarged in the STZ-15 group compared with control mice (*n* = 4–8/group, F_4,29_ = 4.042, *p* = 0.01, one-way ANOVA with Tukey’s post-hoc comparison) (Figure 4C). Notably, the thickness of theca cell layer significantly increased in STZ-treated groups compared with control mice (*n* = 7–8/group, F_4,34_ = 11.59, *p* < 0.0001, one-way ANOVA with Tukey’s post-hoc comparison) (Figure 4D,E). Thus, low doses of STZ, but not 60 mg/kg, induce a PCOS-like morphology with multi-follicular ovaries in comparison with WT and CON.

### 2.4. STZ Treatment Disrupts Mice Estrous Cycle

Because the mice’s estrous cycle represents its ovulation [29], we checked it for 14 days to verify any effect of STZ on ovary functions. All STZ-treated groups presented irregular estrous cycles (Figure 5A). Compared with the CON group, STZ-15 and STZ-30 groups had abnormal cycles for the most part (Figure 5B). In the case of the STZ-60 group, a normal estrous cycle was nearly absent. Similar to the results of Tesone et al., where diabetic rodents were mostly at diestrus [19], mice in the STZ-60 group were mostly (over 80%) at met- or diestrus stages among total estrous cycles, whereas those in the STZ-15 and STZ-30 groups had a notably prolonged estrus stage (Figure 5C).

### 2.5. Systemic STZ Treatment Enhances Intraovarian PI3K-AKT and IRS-1signals

To explore how STZ affects the ovary, we examined the expression of PI3K-AKT pathway-related proteins from the ovary (Figure 6). The expression of PI3K was suppressed in STZ-treated groups. On the other hand, the expression pattern of phospho-AKT (pAKT) was increased in STZ-treated groups, notably increased from the STZ-15 group. The expression pattern of IRS-1 was significantly decreased in the STZ-60 group compared to STZ-15, -30, WT, and CON groups. Therefore, it is assumed that PI3K-AKT signaling was enhanced in the ovary of STZ-treated groups, and IRS-1 signaling was only decreased in the STZ-60 group.

## 3. Discussion

This study demonstrated that consecutive low doses (15 mg/kg or 30 mg/kg) of STZ induce a PCOS-like phenotype in mice. Based on our results, mice in the STZ-60 group represent a typical diabetes model (>300 mg/dL fasting blood glucose level) and those in the STZ-30 group represent a prediabetes model, whereas the STZ-15 group is not related to any condition of diabetes. However, PCOS features were most prominent in the STZ-15 group, where mild stimulation with STZ successfully generated the key features of PCOS, including hyperandrogenism, multi-follicular ovarian morphology, and irregular estrous cycle in young female mice. We could not observe marked hyperandrogenism or significant PCOS morphology in hyperglycemic mice in the STZ-60 group. As the previous reports suggested [13], STZ has been used to induce experimental type 1 diabetes by reducing β-cell function, rather than inducing insulin resistance directly. So, an increase in HOMA-IR of STZ-60 could be due to chronic reducing production of insulin by damaged β-cell, rather than insulin resistance itself.

We found that PCOS features and hormone profiles in mice treated with low doses of STZ are similar to those of animals treated with DHT or DHEA (i.e., PCOS animal models). For example, treated animals have increased antral and atretic follicle numbers, low LH levels, and similar levels of insulin and E2 compared with the control groups [30,31]. However, unlike animals injected with exogenous androgens, body weight gain and metabolic disturbances were absent in mice treated with STZ at low doses. Although the administration of aromatase inhibitors (e.g., letrozole) in PCOS animal models results in the marked suppression of E2 and elevation of LH levels [32,33].

Unlike humans, the ovary is considered the only source of androgen synthesis in female mice and rats [34]. Thus, elevated testosterone levels would be caused by the direct effect of STZ on ovarian synthesis. Ovarian steroidogenesis is mainly controlled by LH action [28]. In addition, this event is potentiated by insulin and insulin-like growth factor-1 (IGF-1) [35]. In our results, both plasmas LH and insulin levels were not elevated in STZ-treated mice. Thus, to explain the cause of increased testosterone levels, we speculate that the insulin signaling pathway in the ovary is altered by STZ stimulation. The effect of insulin and IGF-1 action on ovarian cells would be mediated by the PI3K-AKT pathway [21,25]. Notably, Lan et al. reported that mice not expressing phosphatase and tensin homolog protein (PTEN) in theca cells have elevated testosterone levels with PCOS morphology, but unchanged LH levels [23]. In this context, STZ is known to disrupt the PI3K-AKT pathway [36,37]. However, we found that a low dose of STZ treatment could upregulate IRS-1 signaling. Therefore, we assume that enhanced PI3K-AKT and IRS-1 signalings in the ovary may be one of the reasons for hyperandrogenism and related PCOS-like features in our model.

Regarding the mechanism of STZ actions to alter molecular pathways as well as steroidogenic functions, oxidative damages generated by STZ may have a role since STZ is an alkylating molecule and a potent reactive oxygen species (ROS) agent presenting genotoxicity [16,38]. In male rodents, a high dose of STZ injections accompanied by diabetes depresses steroidogenic functions [39,40]. In human clinical research, elevated oxidative stress is detected in PCOS women irrelative to weight excess [41,42]. Thus, STZ-induced oxidative stress, not excessive, may be related to PCOS development in our PCOS model.

The low dose of the PCOS animal model has some advantages to study PCOS. First, our model uses widely known and profoundly studied single molecules, STZ. Second, compared to the other existed PCOS animal models, low dose STZ treatment does not manifest obesity or other apparent metabolic disorders. Third, the simple and reproductive method can be utilized without the high cost, long waiting time for development, and complicated laboratory techniques.

Our limitations in this study are that we could not examine the other PCOS-related hormones such as anti-Müllerian hormone (AMH) or follicle-stimulating hormone (FSH) in the view of hormonal axis or neuroendocrine mechanism [43,44]. Increased AMH is observed in PCOS women [45] and prenatal AMH exposure induces PCOS features in rodents [46]. In addition, A. Caldwell et al. reported that neuron-specific KO mice did not manifest PCOS-related features after DHT treatment [47]. So, there may be a chance that systemic STZ injections can affect the neuroendocrine hormonal controls related to ovarian functions. Furthermore, we could not manage to examine other PI3K-AKT pathway-related molecules such as Forkhead box O (FOXO)-1. FOXO-1 is a downstream factor of the PI3K-AKT signal pathway which also regulates ovarian follicular development and atresia [48,49]. Several PCOS animal models show upregulated expression of an activated form of FOXO-1 in the ovary [23,50]. Thus, we expect that enhanced FOXO-1 expression may occur in low-dose STZ-treated mice although we could not examine it.

In summary, multiple administration of STZ at low doses can induce PCOS features in mice. This STZ treatment replicates PCOS-like features in mice that are lean and non-diabetic without using exogenous hormones. We propose that STZ may affect animals’ ovaries directly via alteration of the PI3K-AKT pathway. We hope that further cellular and molecular mechanisms related to PCOS can be revealed using this animal model.

## 4. Materials and Methods

### 4.1. Animals

All experiments involving animals were approved by the Institutional Animal Care and Use Committee at Seoul National University Hospital (SNUH-IACUC). Animals were maintained in a facility accredited by the Association for Assessment and Accreditation of Laboratory Animal Care (AAALAC) International (#001169) following the Guide for the Care and Use of Laboratory Animals (8th edition, National Research Council, 2010). Forty female C57BL/6 mice (3–4 weeks old) were purchased from KOATECH (Pyungtaek, Kyunggi-do, Korea) and were acclimated for at least 7 days before experiments. The housing room was maintained under a 12-h light/dark cycle at constant temperature and humidity. Food and water were supplied ad libitum. The experimental end-point was set at 5 weeks after (STZ) treatments. Eight mice were grouped randomly into five different groups: wild-type (WT), control, STZ-15, STZ-30, and STZ-60. One mouse of the STZ-60 group died during the observation period.

### 4.2. Chemicals and Antibodies

All chemicals were purchased from Sigma-Aldrich (St. Louis, MO, USA) except when stated otherwise.

Rabbit polyclonal anti-AKT antibody (#9272, 1:1000), anti-phospho-AKT (pAKT) antibody (Thr308; #9275, 1:1000), anti-phosphoinositide 3-kinase (PI3K) p85 antibody (#4292, 1:1000), and anti-IRS-1 (insulin receptor substrate 1) antibody (#2382, 1:1000) were purchased from Cell Signaling Technology (Cell Signaling Technology, Inc., Denvers, MA, USA). Goat polyclonal anti-actin antibody (#SC-47778, 1:10000) was purchased from Santa Cruz Biotechnology (Santa Cruz Biotechnology, Inc., Dallas, TX, USA). Goat polyclonal anti-rabbit HRP-conjugated secondary antibody (#ADI-SAB-300-J, 1:2000) was purchased from Enzo Life Science (Enzo Life Sciences, Inc., Farmingdale, NY, USA). Donkey polyclonal anti-goat HRP-conjugated secondary antibody (#A15999, 1:5000) was purchased from Thermo Fisher Scientific (Thermo Fisher Scientific, Waltham, MA, USA).

### 4.3. Streptozotocin Treatment

To each experimental group, STZ (#S0130, Sigma-Aldrich, St. Louis, MO, USA) was administered daily for 5 consecutive days at the dose of 15, 30, or 60 mg/kg by intraperitoneal (i.p.) injection (STZ-15, STZ-30, and STZ-60 groups, respectively). The injections were done at the same time of each day with the same vehicle volume. The vehicle was 0.1 M sodium-citrate buffer, which was administrated for 5 consecutive days to the CON group. Additional sucrose supply was not given after STZ treatments.

### 4.4. Bodyweight, Blood Glucose Level, Insulin Tolerance Tests, and Homeostasis Model Assessment of Insulin Resistance Index

Bodyweight and blood glucose level were measured weekly and after the STZ treatment at the same time of the day. Mice have fasted for 6 h before body weight and blood glucose levels measurements. Blood was obtained from a tail prick and glucose concentration was measured on glucose strips in an Accu-Chek^®^ glucometer (Roche, Basel, Switzerland).

Five weeks after the STZ treatment, mice were fasted for 6 h before a baseline blood glucose reading, followed by an i.p. injection of insulin (0.75 IU/kg body weight; Eli Lilly, Indianapolis, IN, USA) for insulin tolerance tests (ITT). Then, blood glucose was checked at 15, 30, 45, 60, and 90 min after insulin injection [51]. The homeostasis model assessment of insulin resistance (HOMA-IR)-index was calculated as fasting blood glucose level (mg/dL) × fasting insulin level (mU/L)/405 [52].

### 4.5. Estrous Cycle Analysis

Estrous cycle patterns were evaluated using the vaginal smear method, as well as visual inspection, according to Byers et al. [29] Vaginal smears were collected once a day at the same time (14:00~15:00) for 14 consecutive days beginning at 4 weeks and 5 weeks after the end of the STZ treatment. The smear slides were stained using crystal violet solution (#V5265, Sigma-Aldrich) and observed under a light microscope. The stages of the estrous cycle, proestrus, estrus, metestrus, and diestrus, were determined as described by McLean et al. [53].

### 4.6. Hormonal Assays

For hormonal analyses, blood samples were taken from the abdominal aorta at the endpoint of the experiments. Samples were centrifuged and the plasma was used for hormonal analysis. Mice were fasted for at least 6 h before blood collection. Hormones levels were measured using enzyme-linked immunosorbent assay (ELISA) kits: rodent total testosterone (#ADI-900-065, Enzo Life Sciences), mouse insulin (#90080, Crystal Chem., Elk Grove Village, IL, USA), rodent LH (#KA2332, Novus Biologicals, Centennial, CO, USA), and mouse estradiol (E2; #CSB-E05109m, CUSABIO, Houston, TX, USA). Microtiter ELISA plates were read in a microplate spectrophotometer (BioTek, Winooski, VT, USA). All procedures were performed according to manufacturers’ instructions in duplicates.

### 4.7. Histological Analysis

Mice ovaries were excised immediately after deep anesthesia using ketamine (87.5 mg/kg; Yuhan Corporation, Seoul, Korea) and xylazine (12.5 mg/kg; BAYER, Berlin, Germany), weighed, and fixed in 4% paraformaldehyde solution (sc-281692, Santa Cruz Biotechnology). Ovaries were embedded in paraffin and sectioned into 4 μm sections. Then, sections were stained by an Autostainer XL (Leica, Wetzlar, Germany) with hematoxylin and eosin. Integrated histological assessment of ovaries was done analyzing seven to eight serial sections, which were 20–25 μm intervals apart. In an ovary that is over 1000 um in diameter, the section was serially sectioned into 40 slides and all the sections were analyzed. Histological images were taken using the EVOS FL auto2 imaging system (Thermo Scientific, Waltham, MA, USA).

Diameters of ovarian follicles and corpus luteum (CL) were measured at the longest and smallest diameter length of the ovary and the mean diameter was calculated. Antral follicles were counted including small and large antral follicles, but preovulatory or preantral follicles were excluded. Follicles were classified as atretic if follicles showed degenerated oocytes or an atrophied granulosa cell layer [47,54]. Microscopic images were analyzed using Image J software (NIH, Bethesda, MD, USA) and Adobe Photoshop CC (Adobe, San Jose, CA, USA).

### 4.8. Immunoblotting

Ovaries samples were obtained immediately after the onset of anesthesia and frozen in liquid nitrogen for storage. Samples were mechanically homogenized in RIPA buffer (Thermo Scientific) and LDL sample buffer (#B0007, Life Technologies, Carlsbad, CA, USA). Protein concentrations were adjusted after measurement using a protein assay kit (#23240, Thermo Scientific). Then, protein samples were subjected to SDS-PAGE followed by anti-actin, AKT, phospho-AKT, PI3K, IRS-1 immunoblotting. Specific signals were visualized, quantified, and analyzed using the Amersham Imager 600 imaging system (GE Healthcare Life Science, Chicago, IL, USA).

### 4.9. Statistical Analysis

Data analysis was performed using GraphPad Prism software version 7 (GraphPad Software, San Diego, CA, USA) and SPSS software version 8 (SPSS Inc., Chicago, IL, USA). Data are shown as means ± standard error of the mean (SEM). Multiple group comparisons were performed using one- or two-way ANOVA followed by post-hoc Tukey’s test. For all analyses, *p* values below 0.05 were considered to indicate statistically significant differences.

## Figures and Tables

**Figure 1 ijms-22-01299-f001:**
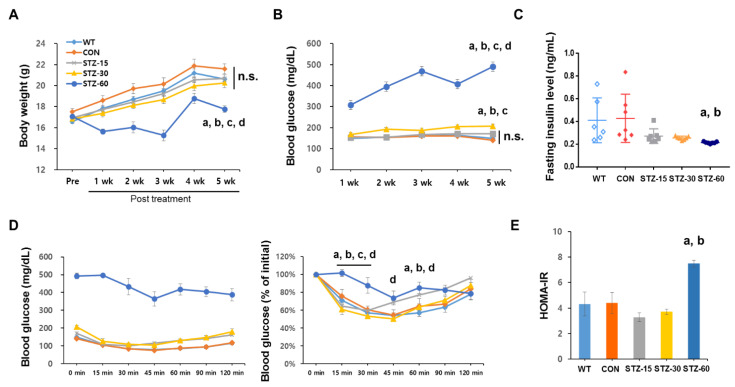
Analyses of bodyweight and glucose level in streptozotocin (STZ)-treated mice. (**A**) Bodyweight of mice in each group during the experimental period: wild-type (WT, diamonds with black dotted line); control (CON, diamonds with a black line); 15 mg/kg streptozotocin (STZ-15, rectangles with bright orange line); 30 mg/kg streptozotocin (STZ-30, triangles with dark orange line); and 60 mg/kg streptozotocin (STZ-60, circles with red line) groups. (**B**) Blood glucose levels in each group. The injection of 60 mg/kg STZ induced a typical diabetes condition whereas lower doses of STZ injections did not. (**C**) Plasma insulin levels: WT, white diamonds; CON, black diamonds; STZ-15, bright orange rectangles; STZ-30, dark orange triangles; and STZ-60, red circles. Treatment with STZ injections did not significantly suppress insulin levels in STZ-15 and STZ-30 groups. (**D**) Insulin tolerance tests (ITT) were performed at 5 weeks after STZ treatment (left) and expressed as a ratio compared to initial values (right). Mice in the STZ-60 showed signs of insulin resistance. (**E**) Calculated homeostasis model assessment of insulin resistance (HOMA-IR) index: WT, white bar; CON, black bar; STZ-15, orange bar; STZ-30, dashed orange bar; and STZ-60, red bar. Insulin resistance was not observed in STZ-15 and STZ-30 groups. (a) significant differences compared with the WT group; (b) significant differences compared with the CON group; (c) significant differences compared with the STZ-15 group; (d) significant differences compared with the STZ-30 group; n.s., not significant.

**Figure 2 ijms-22-01299-f002:**
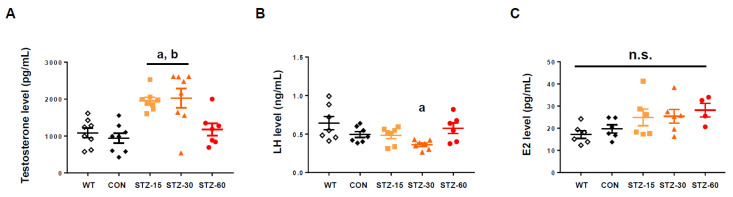
Hormonal assay for testosterone. (**A**) Plasma total testosterone levels in wild-type (WT, white diamonds); control (CON, black diamonds); 15 mg/kg streptozotocin (STZ-15, bright orange rectangles); 30 mg/kg streptozotocin (STZ-30, dark orange triangles); and 60 mg/kg streptozotocin (STZ-60, red circles) groups. Mice in the STZ-15 and STZ-30 groups developed hyperandrogenism. (**B**) Plasma luteinizing hormone (LH) levels. Treatment with STZ injections did not elevate circulating LH levels. (**C**) Estradiol (E2) levels. Treatment with STZ injections did not suppress E2 synthesis. (a) significant differences compared with the WT group; (b) significant differences compared with the CON group. n.s., not significant.

**Figure 3 ijms-22-01299-f003:**
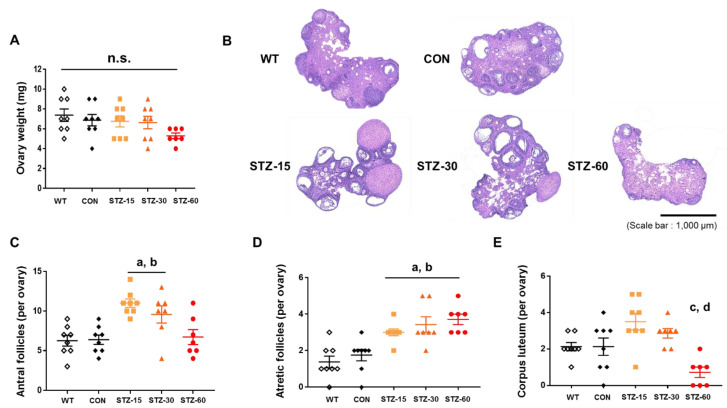
Polycystic ovary syndrome (PCOS) phenotype of STZ-treated mice ovary. (**A**) Ovary weight in wild-type (WT, white diamonds); control (CON, black diamonds); 15 mg/kg streptozotocin (STZ-15, bright orange rectangles); 30 mg/kg streptozotocin (STZ-30, dark orange triangles); and 60 mg/kg streptozotocin (STZ-60, red circles) groups. (**B**) Representative microscopic images of ovary sections (hematoxylin and eosin staining; scale bar = 1000 μm). (**C**) The number of antral follicles per ovary indicating multi follicular formation in the ovaries of mice in the STZ-15 and STZ-30 groups. (**D**) The number of atretic follicles per ovary indicating that STZ injections compromised ovarian functions. (**E**) The number of corpora lutea per ovary showing the decreased abundance of corpora lutea in the STZ-60 group. (a) significant differences compared with the WT group; (b) significant differences compared with the CON group; (c) significant differences compared with the STZ-15 group; (d) significant differences compared with the STZ-30 group; n.s., not significant.

**Figure 4 ijms-22-01299-f004:**
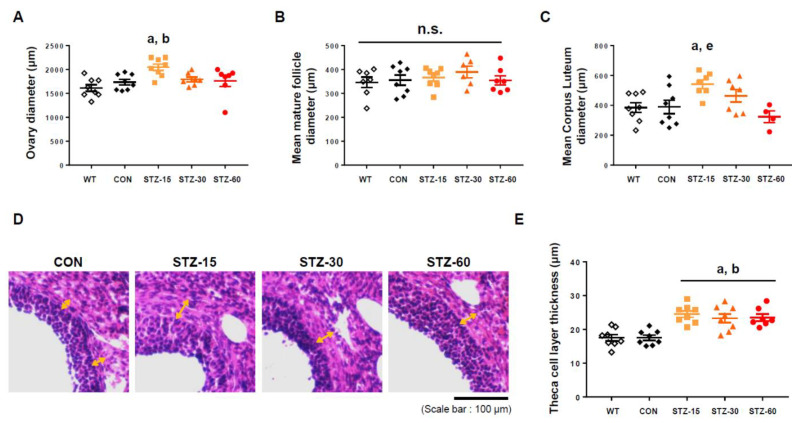
PCOS phenotype of STZ-treated mice ovarian follicles. (**A**) Diameter of ovaries in wild-type (WT, white diamonds); control (CON, black diamonds); 15 mg/kg streptozotocin (STZ-15, bright orange rectangles); 30 mg/kg streptozotocin (STZ-30, dark orange triangles); and 60 mg/kg streptozotocin (STZ-60, red circles) groups. (**B**) The average diameter of mature follicles. (**C**) Average corpus luteum (CL) diameter. Corpora lutea were enlarged in the STZ-15 group. (**D**) Representative microscopic images showing the theca cell layer of antral follicles in each group (hematoxylin and eosin staining; scale bar = 100 μm). (**E**) Theca cell layer thickness indicating significant changes induced by STZ injections. (a) significant differences compared with the WT group; (b) significant differences compared with the CON group; (e) significant differences compared with the STZ-60 group; n.s., not significant.

**Figure 5 ijms-22-01299-f005:**
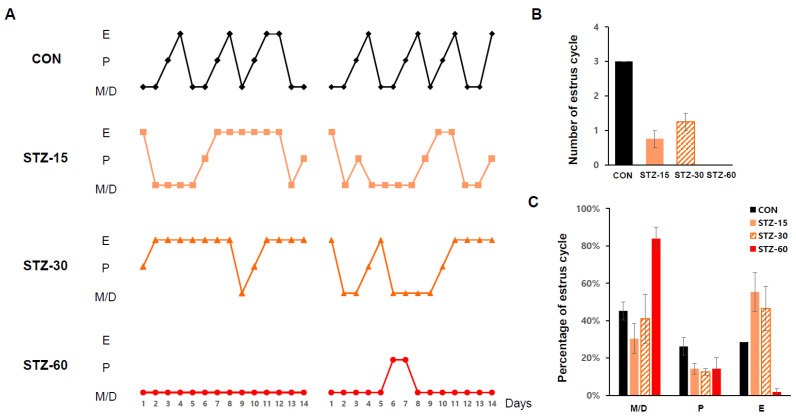
Alterations of estrous cycles in STZ-treated mice. (**A**) Estrous cycle stage classification of each group during 14 days in wild-type (WT, white diamonds); control (CON, black diamonds); 15 mg/kg streptozotocin (STZ-15, bright orange rectangles); 30 mg/kg streptozotocin (STZ-30, dark orange triangles); and 60 mg/kg streptozotocin (STZ-60, red circles) groups. Treatment with STZ injections disrupted ovarian functions. (**B**) The number of normal estrous cycles in each group during 2 weeks in CON, black bar; STZ-15, orange bar; STZ-30, dashed orange bar; and STZ-60, red bar, groups. (**C**) The number of each estrous cycle (%) stage during 14 days indicating that STZ injections alter regular ovary functions in mice. E, Estrus; P, Proestrus; M/D, Metestrus or diestrus.

**Figure 6 ijms-22-01299-f006:**
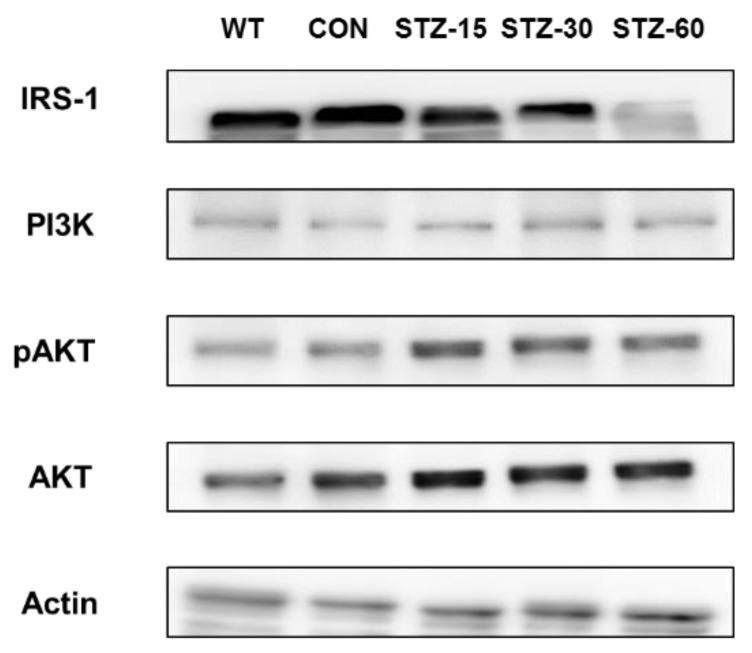
Changes in the IRS-1 and PI3K-AKT pathway in STZ-treated ovary. The expression of specific proteins in STZ-treated ovary. IRS-1 and pAKT were up-regulated along with the STZ treatment dose in the ovary.

## Data Availability

The data that support the findings of this study are available from the corresponding author upon reasonable request.

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
