# Peer review of "Consecutive Low Doses of Streptozotocin Induce Polycystic Ovary Syndrome Features in Mice"

_ijms, 2021, doi:10.3390/ijms22031299_

Round 1

Reviewer 1 Report

This animal study investigates the potential value of using low dose  Streptozotocin (STZ) to develop a mouse model of PCOS. The study found that 5-week treatment with low dose STZ can induce all hormonal and ovarian morphological features of PCOS without induction of any metabolic abnormalities. The authors claim that this method offers several advantages over other mice models such as the simplicity and relatively shorter duration of this method compared to others.

The study is very well-conducted, and the manuscript is well-presented and interesting to read. The results and graphs are clear and easy to follow.

I have the following minor comments to be considered:

  1. Lines 30, 52, 73, 228, 259: Using the term “PCOS symptoms” throughout the manuscript is more of a clinical term. I would suggest PCOS features instead.
  2. Similarly, in line 64, please change PCOS-related conditions to PCOS related features
  3. The term PCOS patients: as PCOS is considered an ovarian dysfunction rather than a disease, it may be better to use the term PCOS women instead.
  4. Line 91: However, mice in the STZ-15 group did not show significantly elevated blood glucose – should be … did not show any significantly elevated blood glucose….
  5. Line 95: it is expected that insulin resistance is associated with high fasting insulin, so the finding of low fasting insulin in the ST-60 does not confirm insulin resistance. While In Fig 1, E, HOMA-IR test shows insulin resistance in the STZ-60 group. There seems to be a discrepancy between these two findings.
  6. Fig 1: Colour codes for STZ-15 and STZ-30 are very similar and difficult to distinguish from each other. Please consider changing one of them to another colour such as blue or green.
  7. Line 171: please change PCO-like condition to PCO-like morphology.
  8. Lines 209 – 218: it would have been better to further test the role of PI3K pathway by using PI3K inhibitor to see if this will stop the effect of STZ on induction of PCOS features.

Reviewer 2 Report

In this manuscript Ryu and colleagues present a novel animal model of PCOS using low doses of streptozotocin. This work is very interesting and could provide a cost-effective method for studying this disease, however, there are several issues that first need to be addressed.

Major

lines 53-5: This statement should be less antagonistic; PCOS is a human disease with as-yet unknown mechanistic causes. I appreciate the authors' model but it is hardly definitive and others cited have improved our understanding of the pathology.

lines 96-7: The authors report no significant difference in fasting insulin between STZ-15/STZ-30 and controls, yet list F and P values that are in fact significant at P < .05 (line 390). Indeed, all STZ groups in Fig. 1C do appear to have lower fasting insulin than controls (which have an unusually large s.d.) This needs to be corrected. Further, the authors conclusions about this model need to be reconsidered (e.g., line 227, line 249) as this data suggests at least a "pre-diabetic" state.

line 161: I have a concern with how diameter is used in this context. The methods state that only the longest length was measured so ovaries could in theory be getting smaller, just longer and thinner, rather than larger as is suggested. Please consider using the largest and smallest diameter to get an accurate measure of the ovary size (as an approximate oval), or clearly state what was measured in the body text.

Fig. 3: Follicles counted are shown as "per ovary", yet the methods (lines 366-8) state that 7-8 sections 20-25 um apart were measured. In an ovary that is over 1,000 um in diameter (Fig. 3B) I am not sure how all follicles could be counted. Please provide further details.

Figure 5: No WT data is shown, despite being presented in other figures and listed in the figure legend. Further, this figure and Fig. 6 are the only figures to be referenced in the body text at all. Please make sure that all figures are appropriately referenced in the Results section.

Figure 6: I appreciate the authors' attempts to look at mechanism, but there are several issues with how this series of experiments were carried out. The methods state that bands were quantified to generate Fig. 6B. However, given the substantial differences in loading (i.e., actin signal) this is not appropriate and band densities should be normalized against the actin band. I'm also not sure where the error bars come from as only one blot per protein was included in the supplementary materials and neither the methods nor figure legend give information as to replicates. The immunoblot for IRS-1 was not run far enough to give reliable results. It appears that the STZ-15 group could have simply been cut off rather than poorly expressed. As it stands, I don't think Fig. 6 or the conclusions supported by this data (lines 256-7) are appropriate. Please consider additional immunoblotting, the possibility that decreased insulin (Fig. 1C) could be the cause of the observed phenotype, or further explore the role of ROS.

Minor

lines 13-4: The way this is phrased makes it seem as though this work can not be used to study mechanisms.

lines 16-7: Consider removing "groups" and simply state the STZ treatments.

lines 30-1: Please rephrase to be more specific; it doesn't seem possible to have PCOS symptoms without metabolic abnormalities. 

line 62: "hypophysial-pituitary" is redundant as hypophyseal pertains to the pituitary gland. Perhaps the authors meant "hypothalamic"?

line 98: ITT is not defined (and may not be immediately understood by the target audience).

line 132: I don't think E2 conversion can be ruled out since the STZ groups do have higher E2 serum levels (although not significant) and no cytochrome activity was measured (although this is stated in line 241 and needs to be removed).

line 147: the semicolon should be replace with a colon.

line 150: the acronym "PCO" is only used a couple of times and can probably be omitted.

line 154-5: Remove "In the case of CL," and the parentheses. Sentence should begin: "The number of corpora lutea per ovary in ..." 

line 164: Please state the criteria for "mature follicle". Do the authors mean antral follicles?

line 171: "Multi-follicular" ovaries are expected in the mouse. Please clarify what is meant here.

Fig. 5C: Please rephrase this caption: "Occurence of each estrus cycle (%) stage." I think the authors mean the length of time (i.e., days) spent in each phase of the estrus cycle.

line 265: Please give some reference to support this statement or remove it.

lines 306-7: Please state which group this mouse belonged to and cause of death (if known).

line 348: This should have been blinded to prevent unconscious bias.
